Fish predation hinders the success of coral restoration efforts using fragmented massive corals

Koval Gammon gnk2@miami.edu 1
Rivas Nicolas 1
D’Alessandro Martine 1
Hesley Dalton 1
Santos Rolando 2
Lirman Diego 1
1 Rosenstiel School of Marine and Atmospheric Science, University of Miami , Miami , FL , United States of America
2 Department of Earth and Environment, Florida International University , Miami , FL , United States of America
Vergés Adriana
Electronic publication date: 2020 Oct 2
Publication date: 2020
Volume: 8
Electronic Location ID: e9978
Received 2020 Apr 22; Accepted 2020 Aug 26
Copyright: ©2020 Koval et al.
Copyright year: 2020
Copyright holder: Koval et al.
License: This is an open access article distributed under the terms of the Creative Commons Attribution License, which permits unrestricted use, distribution, reproduction and adaptation in any medium and for any purpose provided that it is properly attributed. For attribution, the original author(s), title, publication source (PeerJ) and either DOI or URL of the article must be cited.
License URL: https://creativecommons.org/licenses/by/4.0/

Keywords: Coral restoration, Microfragmentation, Fish predation, Massive corals, Orbicella faveolata, Montastraea cavernosa, Pseudodiploria clivosa, Pseudodiploria strigosa

Funding: NOAA’s Restoration Center award OAA-NMFS-HCPO-2016-2004840 This project was funded by NOAA’s Restoration Center (award OAA-NMFS-HCPO-2016-2004840). The funders had no role in study design, data collection and analysis, decision to publish, or preparation of the manuscript.

==============================
As coral reefs continue to decline globally, coral restoration practitioners have explored various approaches to return coral cover and diversity to decimated reefs. While branching coral species have long been the focus of restoration efforts, the recent development of the microfragmentation coral propagation technique has made it possible to incorporate massive coral species into restoration efforts. Microfragmentation (i.e., the process of cutting large donor colonies into small fragments that grow fast) has yielded promising early results. Still, best practices for outplanting fragmented corals of massive morphologies are continuing to be developed and modified to maximize survivorship. Here, we compared outplant success among four species of massive corals (Orbicella faveolata, Montastraea cavernosa, Pseudodiploria clivosa, and P. strigosa) in Southeast Florida, US. Within the first week following coral deployment, predation impacts by fish on the small (<5 cm2) outplanted colonies resulted in both the complete removal of colonies and significant tissue damage, as evidenced by bite marks. In our study, 8–27% of fragments from four species were removed by fish within one week, with removal rates slowing down over time. Of the corals that remained after one week, over 9% showed signs of fish predation. Our findings showed that predation by corallivorous fish taxa like butterflyfishes (Chaetodontidae), parrotfishes (Scaridae), and damselfishes (Pomacentridae) is a major threat to coral outplants, and that susceptibility varied significantly among coral species and outplanting method. Moreover, we identify factors that reduce predation impacts such as: (1) using cement instead of glue to attach corals, (2) elevating fragments off the substrate, and (3) limiting the amount of skeleton exposed at the time of outplanting. These strategies are essential to maximizing the efficiency of outplanting techniques and enhancing the impact of reef restoration.

Introduction

Coral populations have experienced drastic declines due to a variety of stressors (Gardner et al., 2003; McLean et al., 2016). Globally, increases in ocean temperature and ocean acidification are the most serious threats and can lead to mass coral mortality and reduced calcification rates (Hoey et al., 2016). Rising ocean temperatures have resulted in increased frequency and intensity of bleaching events (Heron et al., 2016; Hughes et al., 2019). Also, ocean acidification has begun to reduce coral calcification rates and cause framework erosion (Muehllehner et al., 2016). The magnitude and rate of coral decline require drastic, large scale actions to curb climate change impacts as well as a suite of local conservation and management measures. Active coral reef restoration has developed as one of the tools available to foster coral recovery and restore the ecosystem services that healthy coral reefs provide (Rinkevich, 2019). The present study focused on South Florida reefs where coral abundance has declined due to the interaction of high and low thermal anomalies (Lirman et al., 2010; Manzello, 2015; Drury et al., 2017), nutrient inputs and algal overgrowth (Lapointe et al., 2019), hurricanes (Lirman & Fong, 1997), sedimentation (Cunning et al., 2019), and coral diseases (Richardson et al., 1998; Precht et al., 2016; Walton, Hayes & Gilliam, 2018).

A popular method of coral reef restoration is coral gardening, where coral stocks propagated through sequential fragmentation within in-water and ex situ nurseries are outplanted in large numbers onto depleted reefs (Rinkevich, 2006). Until recently, restoration programs based on the coral gardening methodology have focused primarily on branching taxa like Acropora due to their rapid growth rates, resilience to fragmentation, pruning vigor, and ease of outplanting (Bowden-Kerby, 2008; Shaish et al., 2010; Lirman et al., 2014; Schopmeyer et al., 2017). While acroporids rapidly enhance the structural complexity of reefs, focusing restoration efforts on single taxa ignores the role that diversity plays in ecosystem function (Brandl et al., 2019) and makes restored communities susceptible to disturbances like diseases or storms that affect branching corals disproportionately. Thus, there is a need to expand our restoration toolbox to include multiple species with different morphologies and life histories (Lustic et al., 2020). The use of massive corals for restoration was initially hindered by the slow growth rates associated with these taxa. However, recent developments in microfragmentation and reskinning techniques (Forsman et al., 2015) that accelerate the growth of massive corals have made it possible to use these reef-building taxa for restoration.

The microfragmentation process involves fragmenting corals with massive morphologies into small (<5 cm2) ramets that consist mostly of living tissue and a limited amount of skeleton (Page, Muller & Vaughan, 2018). These microfragments can be mounted onto various types of substrate (e.g., ceramic plugs, plastic cards, cement pucks) using glue or epoxy and allowed to grow by skirting over the attachment platform before being outplanted. Once fragmented, ramets of Pseudodiploria clivosa and Orbicella faveolata grew up to 48 cm2 and 63 cm2 per month, respectively (Forsman et al., 2015) and were thereby capable of achieving colony sizes within a few months that would otherwise take years to develop after natural recruitment. Moreover, a single parent colony can produce hundreds of ramets available for continued propagation and restoration.

The microfragmentation technique overcomes the slow-growth bottleneck, but methods for outplanting fragmented massive corals onto degraded reefs need to be developed and evaluated to maximize outplant survivorship and success. This is especially relevant in Florida and the Caribbean where the massive coral species used here have been severely impacted by the recent outbreak of stony coral tissue loss disease (SCTLD) (Precht et al., 2016; Alvarez-Filip et al., 2019). The present study is one of the first to record the survivorship of small fragments of four species of massive corals (O. faveolata, Montastraea cavernosa, P. clivosa, and P. strigosa) outplanted onto reefs in Southeast Florida, US. To measure the success of this technique, we: (1) documented survivorship and removal probability of fragments outplanted using different techniques, (2) monitored the impacts of fish predation on newly planted fragments, and (3) evaluated different outplanting techniques that may reduce the impacts of predation (i.e., coral fragment removal and mortality).

Materials & Methods

Coral fragmentation

Colonies used in both experiments were collected from a seawall at Fisher Island, Miami, Florida (25.76°N, 80.14°W; depth = 1.8 m). Each parent colony was cut into small fragments (average size = 4.2 ± 1.9 cm2 (mean ± SD) using a diamond band saw, and fragments were attached to ceramic plugs using super glue as described by Page, Muller & Vaughan (2018). After fragmentation, the height of the fragments ranged from 0.5–1.0 cm. The ceramic plugs with corals were placed on PVC frames and then fixed to coral trees (Nedimyer, Gaines & Roach, 2011) at the University of Miami’s in-water coral nursery (25.69°N, 80.09°W; depth = 9.4 m) where they were allowed to acclimate for 4–6 weeks before outplanting (Fig. 1A). After this recovery period, the fragments were strongly cemented (no corals were dislodged during the transport and outplanting steps) but had not fully skirted tissue onto the ceramic plugs. Due to funding constraints, the parent colonies used in this study were not formally genotyped. Nevertheless, fragments from every parent colony were represented in each reef and treatment and, thus, the results combine the survivorship of corals from three parent colonies per species in experiment one and five parent colonies in experiment two.

Figure 1 Corals outplanted using methods from experiment 1 and examples of fish predation.

(A) Coral fragments deployed at the UM nursery for recovery prior to outplanting, (B) Pseudodiploria clivosa fragment outplanted using a ceramic plug, (C) Orbicella faveolata fragment outplanted using a cement puck, (D) triad of coral fragments outplanted using ceramic plugs showing the complete removal of the coral on the top of the image by fish, (E–F) evidence of fish predation on outplanted fragments as shown by the white skeletal lesions without living tissue.

Outplanting experiment one: assessing outplant survivorship

The first outplanting experiment consisted of four coral species with massive or brain colony morphologies: O. faveolata (listed as threatened under the US Endangered Species Act), P. clivosa, P. strigosa, and M. cavernosa. Fragmented corals were glued onto ceramic plugs using polyurethane waterproof glue (Gorilla Glue) (Figs. 1A, 1B). The plugs with the corals were then mounted into cement pucks with 2-part epoxy putty (AllFix). Finally, these pucks were secured onto the reef using cement (1 part Portland cement, 0.1 part Silica fume), raising corals three cm above the substrate to limit sediment and algal interactions (Fig. 1C). Only corals with healthy (no discoloration or lesions) tissue were used. Corals were outplanted onto three reef sites in Miami, Florida, in June–July 2018: Reef 1 (25.70°N, 80.09°W; depth = 6.0 m), Reef 2 (25.68°N, 80.10°W; 7.5 m), and Reef 3 (25.83°N, 80.11°W; 6.4 m). These reefs have low topography and very low cover of stony corals (<1%, Supplemental Data). Corals were collected and outplanted under Florida’s Fish and Wildlife Commission Permit SAL-19-1794-SCRP. Corals were deployed within replicate grids (from 3 × 3 to 5 × 5 m) whose areas were determined based on substrate availability. Four such grids were deployed in Reef 1, 7 in Reef 2, and 6 in Reef 3. The corals were spaced 40–60 cm apart within each grid to maintain a consistent density of corals across replicate plots and grids were separated by at least 2 m. The coral outplants were placed at least 20 cm away from existing stony and soft corals, sponges, and the zoanthid Palythoa. In total, 53 M. cavernosa, 123 O. faveolata, 80 P. clivosa, and 41 P. strigosa were outplanted among the three reefs, with all 4 species represented in each plot.

For each outplant in this experiment, we documented presence/absence of the coral fragment (Fig. 1D) and prevalence (i.e., proportion of corals by species with signs of predation) of tissue mortality caused by predation on remaining fragments (e.g., missing polyps, feeding scars) (Figs. 1E, 1F) at one week, one month, and six months after deployment. The proportion of corals physically removed by fish predators from their outplanting platforms was compared among coral species, reefs, and time since outplanting using a Generalized Linear Model (GLM) following guidance by Warton & Hui (2011). Here, we used a GLM with a binomial distribution and a logit link function to model the probability of outplant removal. The model incorporated species, reef, and monitoring period (time) as fixed variables. Residuals diagnostics plots were used to test model assumptions, and D-squared values, which indicate the amount of deviance accounted for by the models (i.e., analogous to R2), were used to evaluate the goodness of fit of the selected models. Tukey post hoc tests were used to evaluate pairwise differences among the levels of the categorical variables in the models (species, reefs, time). All statistical analyses were performed in R v3.5.3 (R Core Team, 2017).

Outplanting experiment two: reducing predation impacts

Based on the high level of predation observed during Experiment One, a second study was designed to determine if predation impacts could be minimized through modifications to the outplanting method. This experiment tested the role of the skeletal profile (i.e., the height of the coral fragment) and attachment medium (glue vs cement) on coral removal and predation rates. This experiment used coral fragments (average size = 2.8 ± 6.5 cm2 (mean ± SD)) from five P. clivosa colonies. A high level of predation and abundance of fish predators were recorded at Reef 1 in the first experiment, so this location was chosen as the study site for the second experiment.

Compared to fragments with healed, skirting edges, corals with exposed skeletal walls may provide easier access to the coral tissue and encourage growth of endolithic or turf algae that could attract grazing by fish. Hence, we hypothesized that the exposed skeletal profile (height) and the presence/absence of bare skeleton on the sides of a fragment would influence predation patterns (Figs. 2A–2B). We further hypothesized that the rate of the physical removal of outplanted fragments would be related to the attachment method. To test this, we developed a triangular cement platform (cement “pizza”; Figs. 2C–2D) that used cement (in lieu of glue) to secure corals and allowed the height of the fragments to be adjusted by varying the amount of cement used. Corals were secured to the cement pizzas as four treatments:

Figure 2 Corals ouplanted using methods from experiment 2.

(A) Coral fragment showing exposed skeletal walls, (B) coral fragment with tissue-covered walls, (C) corals outplanted using a cement pizza, (D) individual ceramic plugs, (E) fragments with exposed walls on a cement pizza, (F) fragments with covered walls on a cement pizza, (G) fragments placed flushed within a cement pizza, (H) fragments embedded into a cement pizza.

1. “raised exposed”, with fragments placed on top of the cement so that vertical walls (devoid of tissue) protruded from the cement treatment (Fig. 2E);

2. “raised covered”, with fragments placed on top of the cement so that vertical walls (covered with tissue) protruded from the cement treatment (Fig. 2F);

3. “flushed”, with fragments embedded into the cement so that the fragment was level with the cement platform and only the surface of the coral was visible (Fig. 2G);

4. “embedded”, with fragments embedded into the cement so that the coral was positioned one cm below the surface of the cement to prevent access by fish (Fig. 2H).

Individual coral fragments were attached in groups of three (triads) onto each cement pizza to foster coral fusion and faster colony development as described in Paget et al. (2018). The pizzas were cemented individually onto the reef pavement within plots (n = 120 corals placed onto 40 pizzas). Each plot consisted of 10 pizzas (n = 3–4 pizzas per treatment), with each pizza spaced 30–50 cm apart. Plots were separated by 1 m. In addition to using the cement pizzas, coral fragments were mounted onto ceramic plugs using glue and outplanted directly onto the reef (as used in Page, Muller & Vaughan, 2018) to serve as controls (Figs. 1B, 1D). All fragments used as controls had healed skeletal walls covered in tissue. Control corals mounted on plugs were grouped as triads with spacing between corals similar to the pizzas, and deployed directly onto the substrate within the same plots as the experimental corals using cement. Each plot received 5 control triads (n = 20 triads, 60 corals). All corals were outplanted in August 2019 and coral condition surveys were conducted immediately after deployment and again after one and three weeks to document the presence/absence of coral fragments and evidence of tissue mortality caused by predation. The average percent tissue removal was estimated visually for each coral using 10%-classification bins. Lastly, the proportion of the tissue area covered by sediments for each coral outplant was visually evaluated at one and three weeks using the methods just described. Values for these two metrics were averaged within pizzas/triads and compared among treatments using ANOVA.

Coral cover, fish abundance, and predation surveys

The percent cover of stony corals at the three reefs selected was calculated using the point-count method as described by Lirman et al. (2007). At each reef, three plots (10 m in diameter) were haphazardly selected in the vicinity of where the coral fragments were deployed. Within each plot, 25 images were haphazardly collected at a distance of 50 cm from the bottom. The cover of stony coral was calculated using 25 random points overlaid onto each image using the Coral Point Count with Excel extension (CPCe) software (Kohler & Gill, 2006). The proportion of random points placed over stony corals was divided by the total number of points (n = 25 per image) to calculate the proportional cover of corals. Mean percent coral cover was calculated for each plot (n = 3 plots per reef) and averaged for each reef.

Fish surveys to compare fish abundance at coral outplant sites were conducted as part of experiment one at each reef site using the Reef Visual Census (RVC) method (Bohnsack & Bannerot, 1986). Using this method, the surveyor recorded the abundance of fish taxa from a stationary point at the center of the study site within a cylindrical 15-m diameter survey area, extending from the substrate to the surface of the water column. Each survey was completed in 15 min and all fishes observed were identified to species level. Between May 2018 and February 2019, we completed 13 RVC surveys at Reef 1, 10 surveys at Reef 2, and 14 surveys at Reef 3. During the 10-month monitoring period, all three reefs were surveyed within one month, with Reefs 1 and 3 surveyed opportunistically additional times. All surveys were completed by a single, expert observer. The mean abundance of all corallivorous or predatory fish (Robertson, 1970; Randall, 1974) was compared among reefs using ANOVA.

In addition to the visual fish surveys conducted during the coral deployment for experiment one, we deployed a video camera immediately after each coral deployment focused on the newly outplanted corals to document the fish species observed interacting with the corals after the divers had left the plot for both experiments. Each video was viewed and a list of species interacting with the outplants was compiled. The duration of these deployments was variable (1–7 hrs) based on the time spent by the divers at the site during deployment and was only intended to compile a list of fish species approaching and/or biting the corals.

Results

Outplanting experiment one: assessing predation impacts

Two types of fish-predation impacts were documented: (1) physical removal of outplanted fragments, and (2) tissue removal from corals that remained attached to outplanting platforms. The probability of outplant removal was explained by coral species, reef, and time as fixed effects (GLM χ2-test, p < 0.05) and the model explained 67% of the deviance (Fig. 3, Tables S1, S2). One week after deployment, 8% of M. cavernosa (n = 53 fragments outplanted), 12% of O. faveolata (n = 123), 23% of P. clivosa (n = 80), and 27% of P. strigosa (n = 41) fragments were physically removed from the outplant platforms by fish (all sites combined) (Fig. 4A). The ranking of the probability of removal for the four species was consistent across reefs and time (Fig. 3). There was a minor, but significant increase in the probability of removal over time (Fig. 3, Table S2). The majority of removal occurred during the first week, but corals continued to be removed over time, with an additional 1.9% of M. cavernosa, 6.6% of O. faveolata, 7.1% of P. clivosa, and 7.0% of P. strigosa removed between one and six months after deployment (Fig. 4A).

Figure 3 Probability of coral removal by fish based on species, reefs, and time since outplanting estimated using a binomial GLM.

Bars indicate the GLM fitted values (center lines) and 95 percent confidence intervals (upper/lower extension of the box). According to Tukey pairwise tests: Mcav = Ofav ⁄ = Pcliv = Pstri; Reef 1 = Reef 2, Reef 1 ⁄ = Reef 3, Reef 2 ⁄ = Reef 3. Mcav = Montastraea cavernosa, Ofav = Orbicella faveolata, Pcliv = Pseudodiploria clivosa, Pstri = Pseudodiploria strigosa.

Figure 4 Impacts of fish predation on coral outplants.

(A) Average cumulative (±S.D.) percent of corals removed by species at the different survey time points for all reefs combined in experiment 1, (B) Average (±S.D.) percent of remaining corals by species showing signs of predation for all reefs combined, (C) Cumulative percent of outplanted corals removed (left) and percent of coral remaining showing signs of predation (right) for all species combined, (D) Average (±S.D.) abundance of fish taxa that interacted with outplanted corals at the outplanting reefs based on the RVC fish surveys conducted at all three reefs. Mcav = Montastraea cavernosa, Ofav = Orbicella faveolata, Pcliv = Pseudodiploria clivosa, Pstri = Pseudodiploria strigosa.

The species with the highest prevalence of fish bites one week after deployment were the two Pseudodiplora species, followed by O. faveolata. M. cavernosa was the only species that did not show any signs of predation on remaining corals after one week (Fig. 4B). Similar to the rate of removal, predation prevalence slowed over time, as only an average of 0.3% of surviving corals of all four species combined showed fish bites at the one-month survey compared to 9.2% after the first week. After six months, no signs of predation were observed for surviving M. cavernosa and P. strigosa, and <1% of colonies of the remaining two species showed evidence of fish bites (Fig. 4B).

Cover of stony corals recorded at the three outplant sites was very low. Mean coral cover was 0.85 (±1.0) for Reef 1, 0.8 (±0.4) for Reef 2, and 0.04 (±0.07) for Reef 3. The prevalence of fish predation, including complete fragment removal and fish bites, was highest at Reefs 1 and 2, which coincided with the significantly greater abundance of corallivorous fish taxa recorded at these two sites compared to Reef 3 (ANOVA; Tukey–Kramer HSD test; p ≤ 0.05) (Figs. 4C, 4D). The average number of fish observed interacting with the coral outplants (i.e., parrotfishes, damselfishes, butterflyfishes, surgeonfishes, triggerfishes) was 2.7 individuals survey−1 ± 6.0 (mean ± SD) at Reef 1, 2.0 ±4.7 at Reef 2, and only 0.8 ± 2.3 at Reef 3 (Fig. 4B). Complete coral removal was 17% at Reef 1, 26% at Reef 2, and only 7% at Reef 3 after one week (Fig. 4C). Similarly, signs of fish predation were higher among the remaining corals at Reef 1 (13.7% corals with evidence of predation) and Reef 2 (13.1%), while no evidence of fish bites was observed at Reef 3 after one week (Fig. 4C). The fish taxa observed biting coral fragments included butterflyfishes, parrotfishes, and damselfishes (Table S3). Wrasses and surgeonfishes were also observed approaching the coral outplants but not necessarily biting the coral tissue. While no direct evidence of predation by triggerfishes (a known coral predator) was captured, this taxon was seen in the vicinity of outplants in the video collected in experiment two. Grunts, surgeonfish, and wrasses were the most consistently sighted fish across all 3 sites and were recorded during all 37 surveys. Parrotfishes and damselfishes were also regularly observed at all three outplant locations, having been recorded as being present within 34 and 35, respectively, out of the 37 surveys conducted. Chaetodontidae were recorded within 27 of the 37 surveys, and were present during all 13 surveys at Reef 1, 8 out of 10 surveys conducted at Reef 2, but only 6 out of 14 surveys completed at Reef 3. It is important to note that no evidence of fish removing the corals from their outplanting platforms was captured in our video surveys. Nevertheless, the removal by fish was considered the only driver of the missing corals as corals did not detach during transport nor during their time at the nursery where no fish predators are observed (pers. obs.)

In addition to fragment removal and partial tissue mortality caused by predation, complete coral mortality was observed. After one week, 4% of P. clivosa, 5% of O. faveolata, 9% of M. cavernosa, and 17% of P. strigosa fragments that remained attached to the outplant platforms showed 100% tissue mortality (all sites combined). After six months, the cumulative prevalence of complete mortality was 4% for P. clivosa, 16% for M. cavernosa, 27% for P. strigosa, and 41% for O. faveolata fragments. When removal and complete tissue mortality were combined for all corals and sites combined, 26% of corals died after one week, 30% of corals died after one month, and 51% of corals died within six months of outplanting. Overall, M. cavernosa suffered 26% losses (removal + 100% tissue mortality), followed by P. clivosa (40%), O. faveolata (62%), and, finally, by P. strigosa (73%). While it was not possible to ascertain the cause of mortality (besides that visibly caused by predation) among the coral outplants, no evidence of active stony coral tissue loss disease (SCTLD), which affected the reefs of South Florida in recent years, was observed on outplanted or wild corals at any of the sites during either experiment.

Outplanting experiment two: reducing predation impacts

The mode of attachment of outplanted corals influenced removal patterns. After one week, 14% of the corals fixed to ceramic plugs using glue were removed, while none of the corals outplanted using cement within the pizzas were missing. After three weeks, still none of the corals deployed on pizzas were removed, whereas 54% of the corals outplanted using plugs were missing. While none of the corals in any of the four cement pizza treatments were removed, fish predation impacts were significantly affected by coral treatment within the cement bases (ANOVA, p < 0.05) (Fig. 5). No significant influence of plot was documented and data wer thus grouped for all plots. After three weeks, the average percentage of tissue removed by predation was significantly lowest for corals within the “embedded” treatment and highest for corals placed within the “raised exposed” treatment and corals outplanted using plugs (Tukey-Kramer HSD test; p < 0.05) (Fig. 5). No significant differences were found between corals in the “raised covered” and “flushed” treatments (Fig. 5). Predation impacts were lowest among embedded corals, but only corals within this treatment experienced sediment accumulation on the surface of the colony. For corals within the embedded treatment, the average of the total surface area of the coral outplants covered by sediments was 3.1% ± 2.4 (mean ± SD) after one week and 3.8% ± 3.5 (mean ± SD) after three weeks. Neither controls outplanted using plugs nor corals within the other three pizza treatments accumulated sediments on the coral surfaces.

Figure 5 Average percent tissue (±S.D.) surface area removed from Pseudodiploria clivosa outplants based on outplanting method after 3 weeks.

Black bars show fragments outplanted using cement pizzas (n = 10 pizzas per treatment), grey bar shows corals glued onto ceramic plugs (n = 14 triads). Capital letters indicate assignment into significant groupings based on a Tukey-Kramer HSD test (p < 0.05).

Discussion

The use of fragmented massive corals expands the number of coral species available for reef restoration beyond the initial, decade-long focus on branching corals. Massive corals are key reef-building taxa that have experienced accelerated losses in the past few years due to the stony coral tissue loss disease (SCTLD) epidemic that was first detected in Florida in 2014 (Precht et al., 2016; Walton, Hayes & Gilliam, 2018) and has now been documented in several locations in the Caribbean (Alvarez-Filip et al., 2019). The impacts of SCTLD, added to the historical declines in these taxa, has created a need to move from single-taxa restoration to a community-based approach that includes corals with different life histories and disturbance responses (Lustic et al., 2020). While massive corals can be successfully propagated both in situ and ex situ (Becker & Mueller, 2001; Forsman et al., 2015; Page, Muller & Vaughan, 2018), our study identified a significant bottleneck in restoration success caused by fish predation on newly outplanted fragments. In our study, 8–27% of fragments from four species (O. faveolata, M. cavernosa, P. clivosa, P. strigosa) outplanted onto three reefs in Miami, Florida, US, were removed by fish within one week. A prior study from Florida also documented large predation impacts on M. cavernosa and O. faveolata, with 45% and 22% of fragments affected by predation respectively within the first week (Page, Muller & Vaughan, 2018). With coral cover being so low presently on Florida reefs (<1% coral cover on the reefs used in this study), it is likely that fish predation is being concentrated on outplanted corals, posing concerns for restoration in depleted systems until a critical abundance threshold is reached (Schopmeyer & Lirman, 2015). Supporting this concept, Jayewardene, Donahue & Birkel (2009) found lower prevalence of fish bites on coral nubbins in plots with higher coral cover. The concentration of predation on surviving corals after major declines in abundance due to a hurricane was previously documented by Knowlton, Lang & Keller (1990).

Previously, research efforts have focused mainly on the impacts of reef fishes on the abundance and distribution of macroalgae, so our understanding of their direct effects on stony corals is comparatively more limited in the Caribbean. Only 10 families of fishes have been reported to consume coral polyps and even fewer taxa classified as obligate corallivores (Robertson, 1970; Randall, 1974). Species within the Chaetodontidae (butterflyfishes), Balistidae (triggerfishes), and Tetraodontidae (pufferfishes) families are among the most common corallivorous fishes (Hixon, 1997). In this study, butterflyfish, wrasses, parrotfish, surgeonfish, and damselfish consumed or interacted with newly outplanted corals. Except for butterflyfishes that were observed biting coral tissue, it remained unclear from our visual and video observations whether fragments were physically removed by fish grazing on algae growing on exposed coral skeletons or direct consumption of coral tissue.

The high impacts recorded here on coral outplants may be the result of consumptive or territorial activity (or a combination of both). Both butterflyfishes (Reese, 1989; Roberts & Ormond, 1992) and the adult terminal phase male stoplight parrotfish Sparisoma viride (Bruggemann, Kuyper & Breeman, 1994; Bruckner, Bruckner & Sollins, 2000) have been observed to bite corals within their territories, which supports that certain fish species may selectively target new coral outplants as soon as they appear within their territories. Predation impacts on coral outplants were highest within the first week and tapered off with time, declining to <1% of remaining corals removed after six months, suggesting outplant habituation of the fish fauna to new coral “recruits” may play a role. Similar patterns of temporal predation on coral outplants were reported in Guam, where predation impacts from butterflyfishes and triggerfishes were high within one week of deployment (Neudecker, 1979). Whether the decline in coral removal by fish was a result of corals reaching a size refuge as they grew or due to habituation of the fish to the presence of these corals could not be ascertained in this study. The territories of potential fish predators like parrotfishes were not assessed in this study and we were thus unable to differentiate between the impacts of both these factors nor their interaction. Similarly, the high level of predation may have been caused by the relatively close spacing of outplanted corals (30–60 cm) so that once a prey item was detected, detection of additional corals within a grid was autocorrelated. Thus, the potential role of fish territoriality and spacing of corals on impacts on newly outplanted corals needs further investigation, especially considering the high impacts recorded here that represent a drain in restoration resources.

In our study, impacts of predation varied by species, with P. clivosa and P. strigosa experiencing the highest levels of mortality. While the potential reasons for the differences in species susceptibility to predation were not measured here, factors such as palatability, nutritional content, or skeletal characteristics may play a role and need to be investigated further. Nevertheless, prey selection based on coral species has been previously documented for Chaetodon unimaculatus that showed a preference for feeding on Montipora verrucosa in Hawaii (Cox, 1986), by Balistapus undulatus that targeted Pocillopora damicornis over Seriatopora hystrix (Gibbs & Hay, 2015), and by butterflyfish that target Acropora over other coral taxa (Berumen, Pratchett & McCormick, 2005). Similarly, wild and outplanted A. cervicornis and O. annularis were targeted by the territorial three-spot damselfish (Kaufman, 1977; Knowlton, Lang & Keller, 1990; Schopmeyer & Lirman, 2015).

Fish predation impacts varied by reef and were associated with the abundance of fish taxa known to consume coral tissue. Differences in predation impacts on outplanted coral fragments between sites were also documented by Page, Muller & Vaughan (2018) in the Florida Keys. Similar to our findings, Quimpo, Cabaitan & Hoey (2019) suggested that coral outplants were more likely to be detached when outplanted onto reefs with higher biomass of herbivore and corallivore fishes in the Phillipines. Additionally, Quimpo, Cabaitan & Hoey (2019) reported that incidental grazing of herbivorous fish, particularly the parrotfish Chlorurus spilurus, were the main sources of coral detachment, but that the direct predation by corallivorous fishes only minimally affected coral outplants. Incidental fish predation was also observed by herbivorous fish removing algal tissue from nursery ropes in the Seychelles by Frias-Torres & Van de Geer (2015). A simple response to these patterns would be to avoid outplanting on reefs with high abundance of these taxa, but it is important to note that parrotfishes and surgeonfishes (observed here to target outplanted corals) are also key grazers that are essential to maintain a low abundance of macroalgae on reefs (Mumby et al., 2006) and coral nursery settings (Knoester, Murk & Osinga, 2019). Best practices for the selection of outplanting sites developed for Acropora suggest that low abundance of macroalgae is a key attribute of an ideal restoration site (Johnson et al., 2011). In addition to reducing algal overgrowth, damselfish, triggerfishes, puffers, and other corallivorous fish have been documented to limit impacts of corallivorous invertebrates such as the crown-of-thorns starfish (Acanthaster plancii) and Coralliophillia snails (Ormond et al., 1973; Schopmeyer & Lirman, 2015). Hence, avoiding reefs with a high abundance of grazers that also target corals is not a viable option as it may lead to algal overgrowth and higher impacts by non-fish corallivores. There is, thus, a clear need to develop efficient outplanting methods to minimize the impacts of fish predation on reefs with high abundances of fish herbivores.

While fish impacts were the predominant source of physical removal of fragments in the present study, remaining corals experienced tissue losses due to fish predation and other unknown factors resulting in the mortality of >30% of remaining corals after six months. Fish predation has also been shown to reduce growth rates (Meesters, Noordeloos & Bak, 1994), decrease fecundity (Szmant-Froelich, 1985; Rinkevich & Loya, 1989), and increase susceptibility to diseases (Williams & Miller, 2005; Aeby & Santavy, 2006). Mortality of our outplanted corals was much higher than the average mortality (14.8%) reported for A. cervicornis one year after outplanting (Schopmeyer et al., 2017), highlighting a bottleneck that needs to be addressed to optimize the long-term success of using fragmented massive corals for restoration.

Lower fragment removal rates and reduced prevalence of fish predation were related to the attachment method (glue vs. cement). None of the fragments attached by cement were removed by fish predators showing that cement provides a stronger hold for the outplanted corals than the commonly used glue. Higher detachment of coral fragments attached with glue was also documented by Dizon, Edwards & Gomez (2008). Moreover, corals allowed to recover tissue over their exposed skeletal walls prior to outplanting (“raised healed” treatment) had less predation than corals with exposed skeletal walls (“raised exposed” treatment). Colony edges of exposed skeleton can be preferentially targeted by parrotfish feeding on turf or endolithic algae, resulting in the higher prevalence of fish bites recorded (Bruckner & Bruckner, 1998). Thus, allowing fragmented corals to skirt over the exposed skeleton and grow onto the attachment platform would be the desired step before outplanting. This approach is used in the microfragmentation method described by Page, Muller & Vaughan (2018) where small microfragments composed mainly of tissue with limited skeleton are grown ex situ until the coral tissue reaches the edges of the ceramic plug, resulting in larger coral outplants without exposed skeletal walls and low height, thereby reducing predation risk. This would increase the time a fragment remains within nurseries but would also limit predation impacts. Finally, embedding corals into the cement platform (“embedded treatment”) mimics this process by lowering coral height and the amount of skeletal walls exposed and reduced predation prevalence to <1% of corals. While placing corals embedded into cement may be an option for limiting removal and predation mortality, embedded corals had >3% of the coral tissue covered by sediments, highlighting a potential tradeoff between reduced predation and sediment impacts that needs to be further evaluated.

Lastly, one of the factors that may have resulted in the high levels of removal and predation recorded here may have been the size of the coral fragments used in this study. Prior research has shown a relationship between the size of the fragments or colonies used for restoration and their survivorship and susceptibility to fish predation. For example, a size refuge for coral nubbins was documented by Christiansen et al. (2009), Jayewardene, Donahue & Birkel (2009), Gibbs & Hay (2015), and Quimpo, Cabaitan & Hoey (2019) in field or laboratory experiments. Moreover, in a recent study by Lustic et al. (2020), medium-sized (40–130 cm2) colonies of Orbicella faveolata and Montastraea cavernosa outplanted onto a reef in the Florida Keys showed no significant impacts of fish predation, highlighting a potential size threshold for predation-impacts. Thus, the impacts from fish predation may be mitigated in Florida by outplanting larger fragments or, as described by Page, Muller & Vaughan (2018), by deploying smaller corals together as tight clusters to foster fusion and function as a larger skeletal unit. There is a trade-off between the number of corals derived from a single parent and the size of the fragments produced, so controlled experiments on the role of size on predation susceptibility are needed before the optimum size of massive coral outplants can be established, especially in habitats with high levels of fish predation.

Conclusions

As coral declines continue worldwide, active reef restoration has emerged as a powerful management alternative to slow down and eventually help reverse these declines (National Academies of Sciences, Engineering, and Medicine, 2019). As the number of techniques and species used in restoration increase beyond the established success of branching corals, practitioners and scientists are collaborating to develop expanded guidelines and best practices. These are critically needed to both broaden the footprint of restoration while keeping restoration costs down. Our study, based on the restoration of small fragments (<5 cm2) of four species of massive corals, identified predation by fish as a major bottleneck in restoration success as the activities of a subset of fish taxa (butterflyfishes, wrasses, parrotfishes, surgeonfishes, damselfishes) caused both the high rates of removal of fragments and tissue mortality on remaining fragments. Thus, there is a need to develop methods to reduce these predatory impacts for massive-coral fragments for restoration to be an effective tool in Florida. Here, we identified fragment attachment method (cement performed better than glue) and coral placement (fragments performed better with tissue covering the skeletal walls, and deployed either flushed or embedded within outplanting platforms) as factors that can be used to reduce impacts. We also identified the need to conduct additional experiments to discern the interactive role of fish abundance and territoriality on fragment performance and to explore the role of fragment size and species palatability on survivorship. We believe that the recent development and adoption of microfragmentation as a technique for massive coral propagation will provide the corals needed to develop more efficient outplanting methods and circumvent the fish predation bottleneck identified here in the near future, allowing for the successful restoration of these keystone species.

Supplemental Information

Supplemental Information 1 Summary tables of the Binomial Generalized Linear Model used to predict the proportion/probability of coral removal

Shown are the coefficient estimates in relation to a reference point for each factor, standard error of estimates, t statistics, and p values for the null hypothesis of no difference with respect to the reference point. Significant coefficients are bolded. Null deviance, deviance, and D-squared present a quality-of-fit of the model.

Click here for additional data file.

Supplemental Information 2 Tukey pairwise comparisons among categorical variable levels of the Binomial Generalized Linear Model used to test the proportion/probability of fragment removal by fish

Shown are estimates, standard errors (SE), Z statistics, and P values. Significant comparisons are bolded. Mcav = Montastraea cavernosa, Ofav = Orbicella faveolata, Pcliv = Pseudodiploria clivosa, Pstri = Pseudodiploria strigosa.

Click here for additional data file.

Supplemental Information 3 Abundance of fish taxa commonly observed interacting with coral outplants recorded during visual RVC surveys at the three outplanting reefs

Includes scientific and common names as well as average fish abundance with standard deviation for all three outplant sites.

Click here for additional data file.

Supplemental Information 4 Supplemental raw data for both experiments and fish surveys

Raw data

Click here for additional data file.

We would like to thank S Schopmeyer, M Kaufman, J Carrick, J Unsworth, and R Delp for their help in the field. We appreciate the input by PeerJ editors, Dr. E Muller, and two anonymous reviewers.

Additional Information and Declarations

Competing Interests

Author Contributions

Field Study Permissions

Data Availability

The authors declare there are no competing interests.

Gammon Koval and Nicolas Rivas conceived and designed the experiments, performed the experiments, analyzed the data, authored or reviewed drafts of the paper, and approved the final draft.

Martine D’Alessandro conceived and designed the experiments, performed the experiments, analyzed the data, prepared figures and/or tables, authored or reviewed drafts of the paper, and approved the final draft.

Dalton Hesley conceived and designed the experiments, performed the experiments, authored or reviewed drafts of the paper, and approved the final draft.

Rolando Santos analyzed the data, prepared figures and/or tables, authored or reviewed drafts of the paper, and approved the final draft.

Diego Lirman conceived and designed the experiments, performed the experiments, analyzed the data, prepared figures and/or tables, authored or reviewed drafts of the paper, and approved the final draft.

The following information was supplied relating to field study approvals (i.e., approving body and any reference numbers):

Corals were collected and outplanted under Florida’s Fish and Wildlife Commission Permit (SAL-19-1794-SCRP).

The following information was supplied regarding data availability:

The raw data are available in a Supplemental File. Code is available at GitHub: https://github.com/gkoval11/coral_predation_publication/blob/master/coral_predation_code.R.

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
