# Peer review of "Fish predation hinders the success of coral restoration efforts using fragmented massive corals"

_PeerJ, doi:10.7717/peerj.9978_

## Round 0.1 · original submission · Major Revisions

Please find below three thorough and highly constructive reviews of your manuscript. All reviewers agree that this study provides valuable new knowledge that will be of interest to marine ecologists and restoration practitioners. However, they also highlight a number of methodological issues that need to be addressed, and they also provide suggestions to better contextualise this work. Please make sure to address all comments in your revised manuscript. Looking forward to reviewing it again.

Reviewer 1 ·

Basic reporting

Meets all basic reporting requirements.

Experimental design

All areas of the experimental design standards are met, with the exception of sufficiently detailed methodological details for some sections (fish surveys, % area estimates, and others as outlined in the review text). Please see review for more details.

Validity of the findings

No comment

Additional comments

The manuscript by Koval and colleagues presents the results of two field experiments undertaken in Florida, USA, to investigate the effects of fish predation on coral microfragments used in coral reef restoration. Four species of massive corals were generated using an established restoration technique and were outplanted at 3 replicate reef sites. Heavy predation by fish was observed during the first experiment, so a second experiment was designed to test the effectiveness of a variety of deployment techniques in reducing the impacts of predation.

Overall the study was well written, well organized, and was sound in the analyses. I commend the authors on the clear and organized manuscript.

My only concerns about the study are regarding some methodological details that were missing and that require clarification. Please see the specific points below. Additional minor edits and comments can be found in the attached annotated PDF.
• Line 99. More details are needed about the experimental design, spatial arrangement of fragments and sample size, particularly for experiment 1.
o If corals were 30-50 cm apart, I think it is likely that the impacts of predation on one fragment could influence the others in close proximity, which would mean that the individual fragments are not independent and are spatially autocorrelated.
o How many replicate fragments per species per site were deployed? and were there multiple clusters of corals at each site? Were the corals deployed in species-specific clusters, or all mixed together?
• Line 169/Figure 4D. The fish abundance and predation surveys require more methodological details.
o Are the data presented in Figure 4D from the visual surveys only? Or both visual surveys and video?
o 1-7 hours of video is a huge range. Was there a comparable amount of video captured from each site? Were they standardized for the time of day (fish behaviour is often diurnal/crepuscular). It is unclear if any data were collected from the video and used in analyses, and if so, how?
o Was the RVC survey conducted for a set duration? To what taxonomic level were the fish data collected? Were the data collected by one observer or multiple observers (there can be a strong observer bias in fish visual census data collection)? I would recommend at the very least including a supplementary table with the dates and times and other details of each survey done across the reefs, in addition to Supp table 3. I would also suggest adding the sample size (number of surveys), average survey time, and standard deviations around the mean estimates to that table S3.
• Line 102. Please provide more detail to describe where corals were outplanted and add contextual information about the reef. Were the fragments deployed onto cement hardbottom with low coral cover? Amongst or near corals or in areas of high cover? Near conspecifics? Were they near any obvious fish territories? Deliberately placed within or outside of fish territories? Additional contextual information would be particularly helpful for the discussion – are removal rates so high because obligate corallivores have no other food sources due to low coral cover at these sites?
• Line 164, Fig 5. The authors report the % of tissue removed by predation in Figure 5. However, it is unclear how the average percent tissue removed was actually estimated (Line 164). Was the % estimated in situ by eye? Was the estimate made through image analysis of the fragments post-censusing? Was this an estimate on each fragment averaged across pizzas/triads? Please clarify.
• Line 90. Please define the ‘height of skeletal wall’ in more detail. Does this mean the depth of the skeletal structure below the penetration of live tissue? I ask this because the 4 species used are quite different in terms of polyp depth and the depth of live tissue penetrating within the skeleton. This can have follow-on consequences for time required for tissue regeneration on the sides of the fragment.
• Lines 102-103. Please provide details for the epoxy and cement types used.

Annotated reviews are not available for download in order to protect the identity of reviewers who chose to remain anonymous.

Reviewer 2 ·

Basic reporting

The English in the manuscript is good and easily understandable. The introduction is arranged in a logical manner, with emphasis on why the work was done. However, I found that some of the references are old, especially those that pertain to the impact of fishes on coral nubbins. I have listed quite a few papers that could bring more emphasis on the role that fishes play in the detachment or partial mortality of corals (please see GENERAL COMMENTS – Introduction). I suggest that this be mentioned in a paragraph or two in the introduction as it was a major objective of the study.

Sections were appropriately arranged according to the suggested format of the journal and figures were understandable and were of high quality as they had minimal blurring even when inspected up close. There are, however, a few minor details in the captions and images that might be improved upon (please see GENERAL COMMENTS – Figures & Tables). Moreover, the raw data is supplied by the authors and their R code is readily available at GitHub.

Experimental design

The paper fits the scope of the journal as the topic centers around the Biological and Environmental Sciences, is a research article and has scientific and methodological soundness. Elaborating on the latter, I found the paper discussed well the knowledge gap on the fate of outplanted microfragments especially when exposed to fishes. Generally, the research question/hypothesis are well defined, but I found that some of the section placements were a bit odd (please see GENERAL COMMENTS – Methodology).

Research permits were secured prior to the commencement of the experiments/surveys (SAL-19-1794-SCRP) and the methods used were appropriate for the research question investigated (and broadly comparable to published literature). Methods were generally well described and can be easily visualized from reading the text and if supplemented by the figures that the authors provide. However, I have a few clarifications on the number of replicates, the scoring of coral tissue mortality and the variable video duration of the fish surveys (please see GENERAL COMMENTS – Methodology).

Validity of the findings

Data are provided by the authors, together with the R code that was used to run the statistical analyses. I have a few clarifications regarding the statistical analyses, particularly the response variables used in the GLM (i.e. is “proportion removed” (L115) the presence/absence of the microfragment, the tissue mortality or a combination of both?).

The authors did a decent job at addressing their research questions (i.e. examine coral nubbin detachment or mortality due to fish predation, how that varied among coral species, reefs and time periods, and whether outplanting techniques can mitigate detachment/predation). They have discussed in detail that indeed fish can substantially detach or prey on their microfragment corals, with the extent of detachment/predation influenced by the coral species transplanted, the abundance of reef fishes in a reef site and time since first outplanted (i.e. through time detachment/mortality declines). Moreover, they identified that outplanted corals are less susceptible to detachment/predation when cement is used over glue and if corals are allowed to recover tissue.

However, I think that the discussion can be further improved by incorporating more details on fish detachment/predation on coral nubbins, and how the authors’ results (i.e. rates of detachment/mortality) are similar/dissimilar to these other studies (please see GENERAL COMMENTS – Discussion).

There are some speculations in the discussion regarding unexplained outcomes of the experiment (e.g. L324-326 mortality through time with unidentified causes), but the authors have highlighted these sections and mentioned that these will be interesting research avenues in the future.

Additional comments

The authors have presented a generally well written paper that examined the influence of reef fishes on the detachment or predation induced mortality of outplanted microfragment corals. As pointed out, this technique of coral fragmentation is relatively novel and as such sources of detachment/predation and rates of such removal are not well understood. Here, they identified that a substantial percentage (as high as ~ 30%) of outplanted corals are detached/eaten by reef fishes, with the magnitude of detachment/predation varying according to coral species used, reef location and time since initial outplanting. Moreover, they identified outplanting techniques that may reduce detachment/predation impacts of fishes. Overall, the paper improves our understanding of how reef fishes can be detrimental to outplanted corals, particularly for corals that are derived using micofragmented techniques.
However, I find that some of the section arrangement (i.e. L124-167) unorthodox and that the discussion lacks a bit of literature pertaining to the influence of fishes on coral nubbins. I believe that the content of the discussion can be further strengthened if the authors incorporate these previous works.

Introduction
L42-45: These two stressors are likely the two most pressing issues that coral reefs face due to their large spatial coverage (Hoey et al. 2016 – Diversity), and it might be good to highlight that. “Globally, rising ocean temperature and ocean acidification are perhaps the ….. and can lead to mass coral mortality and reduced calcification rates”.
L49-51: Coral restoration is only one of the suites of tools used to conserve coral reefs. It might be good to indicate that addressing these issues (e.g. climate change) will need proper global and local management and governance, but active restoration programs can aid in the faster recovery and rehabilitation of reefs.
Methodology
L70-71: Commonly attached using glue, epoxy, etc.? Kindly indicate
L76-78: I think this section can be expanded by specifying why is there a need to develop and evaluate ways to maximize outplant survivorship and success. For example, outplanted branching corals are vulnerable to detachment by hydrodynamic forces (Shafir et al. 2006 – Mar Biol; Shaish et al. 2008 – J Exp Mar Biol Ecol) or through the predation (of fishes or invertebrates) or incidental grazing of fishes (Miller & Hay 1998 - Oecologia; Christiansen et al. 2009 – Coral Reefs; Jayewardene et al. 2009 – Coral Reefs; Frias-Torres & van de Geer 2015 - PeerJ; Gibbs & Hay 2015 – PeerJ; Gallagher & Doroupolos 2017 – Coral Reefs; Knoester et al. 2019 – Mar Ecol Prog Ser; Quimpo et al. 2019 – Rest Ecol). The latter (i.e. influence of fishes), I think needs to be particularly mentioned since it is a key question in the study.
L95-96: Can you kindly clarify why this is important to highlight?
L98: Can you please mention how many replicates were done per coral species?
L101-104: Sentence can be improved. I suggest “Microfragmented corals … epoxy and were secured to the reef using cement, with 30-50 cm (from L109) separating each coral outplant”
L109: Please remove “The corals were spaced …” if authors’ agree with the changes above
L110: Is tissue mortality here scored as % of tissue loss? Kindly specify.
L115: Is proportion of corals removed referring to detachment/dislodgement, fish predation or cumulatively an effect of both? Kindly specify.
L124-167: It is quite unorthodox to include research questions and hypothesis in the methodology. I suggest some of the concepts, particularly how coral height and method of attachment be discussed in the introduction. These can be incorporated into the section (i.e. L76-78) that discusses how fish influence coral nubbins, since previous studies have shown that coral nubbin size (Christiansen et al. 2009 – Coral Reefs; Jayewardene et al. 2009 – Coral Reefs; Quimpo et al. 2019 – Res Ecol) influences that rate of detachment/mortality in field and laboratory settings. Moreover, method of attachment has also been shown to affect detachment, but the reasons for these were not really identified (e.g. Dizon et al. 2008 – Aquatic Conser; Levy et al. 2010 – Ecol Eng).
L171: Can you kindly expound on why deployment times for the video cameras were variable (i.e. 1 to 7 hours)?
L171-173: Sentence can be improved to explain why the fish surveys were done. “Additionally, …. to examine differences in fish species assemblages at each site …”
L177: Reference used to classify trophic group of fishes (e.g. fishbase)?
Results
L187-190: It might be better to move this sentence down (i.e. after comments on L 192-193).
L191: It might be better to move this up to L187 after “species-dependent”. For example, “The probability …. with clear ranks in coral susceptibility to removal that was consistent among experimental periods and reef locations”.
L192-193: It might be better to move this sentence after the sentence above (i.e. comments on L191). “Moreover, there was a minor, but significant … in removal between … “.
Discussion
L270: It might also be good to highlight that removal declined with time to emphasize that such high rates of removal (8-27%) are only within the first few days after outplanting
L271-272: Maybe highlight results from the authors’ own work here and relate with the previous study by Page et al. (2018). For instance, “Rates of predation were slightly lower at ~ 5-15% (estimated from Fig. 4B), which were x-fold to y-fold lower than predation rates by Page et al. (2018).”
L280-L286: This section can be further improved by incorporating recent research on family-specific consumption/detachment of corals by reef fishes (e.g. Triggerfish: Gibbs & Hay 2015 – PeerJ, Frias-Torres & van de Geer 2015; Butterflyfish: Gallagher & Doropoulos 2015 – Coral Reefs; Quimpo et al. 2019 – Res Ecol; Blennies: Christiansen et al. 2009; Parrotfish and Rabbitfish: Quimpo et al. 2019 – Res Ecol; Boxfish: Jayewardene et al. 2009).
L292: Size refuge for coral nubbin has been demonstrated by Christiansen et al. (2009) Jayewardene et al. (2009), Gibbs & Hay (2015) and Quimpo et al. (2019) in field or laboratory experiments.
L299-302: As butterflyfishes were the only species that were recorded to consume coral tissue in this study, it may be good to expound on the diet of these fishes. Specifically, do they consume any of the four species regularly, and if no studies have been done to understand whether they do, maybe a more general statement about the proportion of massive corals in their diet would suffice. Berumen et al. (2005 – Mar Ecol Prog Ser) and the book by Morgan Pratchett (Biology of Butterflyfishes) are potentially good references.
L299-304: Feeding selectivity has also been demonstrated for Balistapus undulates, wherein they preyed more on nubbins of Pocillopora damicornis over Seriatopora hystrix (Gibbs & Hay 2015).
L308-311: This sentence could be moved to the paragraph before this one (i.e. L299-304) as this paragraph talks about the species responsible for removing coral nubbins.
L311-314: Indeed, grazing by fishes is important to control algae at reefs, recent studies have also shown that in experimental outplants, ~ 25-80% of turf algae are removed by herbivores (i.e. Ctenochaetus striatus, Chlorurus spilurus and Siganus fuscescens) (Knoester et al. 2019 – Mar Ecol Prog Ser; Quimpo et al. 2019 – Res Ecol).
Figures
Fig. 1B – Maybe mention what species of Pseudodiploria since 2 species were used
Fig. 1E, F – It may be better to encircle or point at the tissue lesion as unfamiliar readers may not readily distinguish these feeding scars.
Fig. 5 – Maybe specify that these were P. clivosa fragments as this was the only species used in the second experiment
Tables
STable 1-3: No captions?

·

Basic reporting

The manuscript by Koval et al is well written, uses sufficient references to cover background info (unless identified below), with appropriate figures, data sharing, and hypotheses. It is also timely and tackles an important issue worth studying, restoration best practices.

I outlined aspects that need to be addressed below...

Introduction
Line 49: consider adding in a reference discussing white band or white plague within S. Florida since diseases were decimating the area long before SCTLD.

Line 54: perhaps change to “Until recently, restoration programs…” since microfragmenting massive corals has occurred within the FRT since 2012/2013 and tens of thousands of outplants have already been placed on reefs at this point

Experimental design

Methods:
Line 112: this says that genotypes were not tracked, which suggests that you did not identify which outplant came from which parent colony. Is this the case? Or did you mean to say that genotype was not assessed molecularly, but you tracked the colonies from particular parent colonies? Please clarify.
Line 116: spell out GLM and then abbreviate after; what factors were your predictor variables and random factors?

Line 155: there are substantial differences between your ‘controls’ and the Page et al method including: 1. size of corals, all outplants in Page et al. are grown out on land for 4-12 months after microfragmentation prior to outplanting until they reach the edge of the plugs. See figures in Page et al.
2. There is no raised skeleton on the microfragments, as they consist of an incredibly thin layer of tissue with very little skeleton when they are initially fragged.
3. the ceramic plugs are attached with epoxy to be a gradual slope from the plug to the substrate.
4. no cement is ever used for attachment
5. microfragments are usual created as ~ 1 cm 2 frags, so I would actually not call your frags technically 'microfrags'. Just something again to acknowledge.

This needs to be acknowledged because as written it suggests the Page et al method is your control, but you have not followed the Page et al. methods in two key ways.

Line 165: what’s the difference between ‘pizza’ and ‘plug triad’? I think they are the same so just use one here or explain the difference between the two. Maybe plug triads are for the control group. This just needs maybe one sentence to clarify.
Line 166: how was this visually assessed? Estimate percent covered? Ranked?
Line 170: 1 – 7 hours of a range seems like a lot. Why so variable and how did you account for this in the analyses? How did you tackle the video surveys vs in situ surveys? How did you collect data from the videos?

Any attempt at capturing growth rates over those 6 months to assess not just survival but whether the corals are growing too?

Validity of the findings

Results
Line 184: this needs to be explained in the methods (variables used etc).
Line 185: what does the sequential addition of variables mean? Sounds like you ran the model, then added another factor and re-ran, but then you only provide on p value and R2 value. Does this represent all the variables included in the model?
Line 191: maybe just stay ‘consistent through space and time”
Line 195: is this cumulative or between 1 months and 6 months (5 month time period). Shouldn’t these two time periods be standardized by day so that you are comparing rates rather than two time points that cover vastly different periods of time?
Line 201: provide an average value here rather than >9%, which could be anything.
Lines 205 – 223: I am confused why there are results on fish abundance here. The methods suggest fish data was collected for experiment 2, but these are results of experiment 1. Please clarify in the methods. Also are these results from in situ diver or video methods?
Line 232: What does “After this study” mean? By the end of the study?
Line 250: extra parenthesis here

Additional comments

Discussion
Line 262: use the reference from Mexico (Alvarez-Filip et al in PeerJ) to show that SCTLD is other places besides FL
Line 272: what I think is important to note within the Page et al. paper is that it was site specific. Offshore corals were highly predated while nearshore corals were predated much less. Acknowledging that site specific interactions between outplants and the fish community is another variable that needs to be addressed in the discussion.

Line 296: there is another part of this…grow out period. You can microfrag and then have longer grow out to reach the size needed for better outplant success. Microfragments are typically outplanted at full plug size with very little plug showing. Then the epoxy shores up the side of the disk. So it looks like your methods were not necessarily typical for what has been used as methods for current outplanting efforts. I think this needs to be acknowledged too.

Line 302: make active voice rather than passive voice

Line 305: did you run regression analysis on this? If not, then you should use different words than ‘related’.

Additional thoughts: why is predation such an issue for massive coral species and not Acropora cervicornis when they are both outplanted within fish territories and on reefs with high abundance? Might be worth discussing here as well.

Can you tell from the video if the fish were biting the corals on the cement pizzas but just not removing them? Or were they deterred from biting the substrate in general?

Please acknowledge growth rates and other ramifications for the methods you are suggesting. Although cement may decrease loss from predation, what could be the unintended consequences such as increased time for outplanting coral, potential influence on reduced growth rates by using cement (especially corals embedded in cement…they would take forever to fuse)?

I know you guys use cement all the time in restoration, but we see a significant reduction in growth rates for corals within our ex situ system that grow on cement vs ceramic plugs. The publication is in prep right now. We had to switch back from using cement to ceramic just to keep growth rates up. I think this at least needs to be discussed as a possible issue.

Conclusion: seems like there are confounding factors as to whether it is ‘glue’ or the ‘ceramic plug’ that may have influenced removal rate. This needs to be clarified throughout the text. It could be the exposure of the plug since you did not grow out the tissue to the edge.

Acknowledgements: suggest identifying role of those you mention, any funding sources needed to be mentioned here?

Figures: recommend changing results presented within the figure legend to a visual within the figure. The equal signs do not show all comparisons (e.g. does Reef 1 = Reef 3?)
Identify ‘coral species’ rather than ‘species’ in legends (as you discuss fish and coral in this paper)

---

## Round 0.2 · Minor Revisions

The two reviewers agree that you have addressed all concerns but also raise some methodological issues that still require clarification - please see below.

Reviewer 1 ·

Basic reporting

No comment

Experimental design

Most methodological details have been clarified and the discussion has been significantly expanded to address the points raised by the reviewers. I have some methodological questions remaining, however, and I suggest they be addressed prior to publication, both for reproducibility and for clarity around the experimental design and analysis.

Validity of the findings

Findings appear clear and valid.

Additional comments

The manuscript by Koval and colleagues presents the results of two field experiments undertaken in Florida, USA, to investigate the effects of fish predation on coral microfragments used in coral reef restoration. Four species of massive corals were generated using an established restoration technique and were outplanted at 3 replicate reef sites. Heavy predation by fish was observed during the first experiment, so a second experiment was designed to test the effectiveness of a variety of deployment techniques in reducing the impacts of predation.

Overall the study was well written and well organized, and the revision made by the authors addressed a majority of my comments satisfactorily. Most methodological details have been clarified and the discussion has been significantly expanded to address the points raised by the reviewers, particularly the additional references and discussion around fish predation patterns, the potential for density-dependent and spatially autocorrelated effects, and key differences with prior outplanting studies (i.e. Page et al.). The findings will be useful to the coral restoration community and both provide guidance for future deployments and identify further research needs.

I have some methodological questions remaining, however, and I suggest they be addressed prior to publication, both for reproducibility and for clarity around the experimental design and analysis. They are outlined below. Please also find additional minor comments/edits in the attached annotated PDF.

- Line 119. Thank you for the additional details around the deployment design for experiment 1. However, please clarify why there are two quadrat sizes indicated (3x3 and 5x5) and how they were distributed among reef sites. I recognize that the size was ‘based on substrate availability’ but I am concerned that differences in spatial distribution among outplants within grids could significantly impact the predation patterns observed. A 25m2 grid is nearly 3 times the area of a 9m2 grid and thus the density of corals could be much different among plots. In line 346 of the discussion, you mention that high predation may have been caused by close spacing of the corals. So, was the spacing of corals within plots consistent across all plots and reef sites (I recognize that you say 40-60 cm spacing)? If densities were consistent among plots, and easy solution would be to add a sentence to the effect of: “corals were deployed within replicate square grids (from 3x3 to 5x5 m) whose areas were determined based on substrate availability. The number of coral outplants placed within each grid ranged from X to X to maintain a consistent density of corals across replicate plots”.

- Line 172: The experimental design is still somewhat unclear for experiment 2. Firstly, if each plot had 10 pizzas total (and 3-4 pizzas per treatment), shouldn't there be at least 12 pizzas per plot given that there are 4 treatments? The numbers don't seem to add up. Secondly, please indicate the size and density (see comment above) of pizzas within each plot, and explicitly state the number of replicate plots (I presume 4 give the sample sizes reported?). Thirdly, I suggest moving lines 177-182 (traids) right after 170 so the 4 experimental treatments can be compared with the control. Then you can discuss how many of each treatment (& control) pizzas/triads were placed in each plot. Lastly, if the pizzas within plots were spaced 30-50 cm apart but plots were only separated by only 1 m, how are the plots independent replicates? Can you provide any reference or justification for the spatial arrangement of the experimental design?

- Line 189: You state that % tissue covered by sediment and % tissue removal were averaged within pizzas and then compared among treatments using ANOVA. But what about the removal data? Were those also averaged within pizzas, or were they combined across all replicate pizzas within a plot? Was replicate pizza or replicate plot used as the experimental unit in the data analysis? More details are needed.

- Line 205: Corals were monitored at 1 wk, 1 mo and 6 mo. But there are many more than 3 RVC surveys on each reef and the number of surveys differed among reefs? Please clarify how the RVC surveys relate to the coral monitoring, how many replicate surveys per time per reef, how many replicate temporal surveys, etc. were undertaken. It is difficult to see how the survey times/dates relate to the coral restoration experimental timeline. The replicate # surveys, and when they were undertaken need to be clearly linked with exp 1. Secondly, given that the sites had different numbers of surveys, should only surveys conducted at all three sites within the same month be used in the analyses? Could there be seasonal variability in the fish communities observed?

General comments:

- With respect to organisation/flow, the methods for the “coral cover, fish abundance and predation surveys” subsection still seemed out of place. It wasn’t clear that the RVC data were used only for experiment 1 but that the video surveys were used for both experiments. At the very least, an introductory sentence in this section is needed to provide context to the data and to explain which experiments each data set was supporting (consider moving line 200 up to the start of the paragraph and expanding it). Also, since the fish survey data are reported in the results along with experiment 1 results, perhaps they should be presented that way in the methods, so that the methods and results are presented in parallel?

Minor comments (also please see attached PDF):

- Typo in TableS2 column title

- Table S3, please add abundance unit to column name and consider adding a SD value for each average. Furthermore, multiple temporal surveys were undertaken. When and how many surveys were completed at each reef that went into the average calculations?

Annotated reviews are not available for download in order to protect the identity of reviewers who chose to remain anonymous.

Reviewer 2 ·

Basic reporting

No Comment

Experimental design

No Comment

Validity of the findings

No Comment

Additional comments

I thank the authors for considering and incorporating the comments/suggestions raised by myself and the other reviewers. The paper has been further polished and I only have a few very minor comments regarding grammar and word usage.

L109: Fragment corals “were” first glued

L149-151: Sentences closely resemble those of L128-131. Perhaps reword to “Funding constraints precluded genotyping of parent colonies, however, every parent colony was represented in each treatment”.

L197: Perhaps spell out Coral Point Count with Excel extension (CPCe) for those unfamiliar.

L223: This could be removed as L221 already mentions the (coral) species-specific removal.

L245-256: I’m hesitant to categorize surgeonfish as “corallivores” since studies have conclusively shown that they are herbivores (e.g. stomach modifications, feeding behavior, etc.; see papers by Howard Choat or David Bellwood). Similarly, damselfish are generally territorial herbivores, planktivores or omnivores.

L270: “and” 41% for O. faveolata

---

## Round 0.3 · accepted · Accept

All the additional comments from reviewers have been adequately addressed. The authors have either resolved or clearly justified all issues raised. Congratulations on a valuable study that should contribute useful information to future coral restoration efforts.